# Catalase (*CAT*) Gene Family in Wheat (*Triticum aestivum* L.): Evolution, Expression Pattern and Function Analysis

**DOI:** 10.3390/ijms23010542

**Published:** 2022-01-04

**Authors:** Yan Zhang, Lanjie Zheng, Liu Yun, Li Ji, Guanhui Li, Manchun Ji, Yong Shi, Xu Zheng

**Affiliations:** State Key Laboratory of Wheat and Maize Crop Science and Center for Crop Genome Engineering, Longzi Lake Campus, College of Agronomy, Henan Agricultural University, Zhengzhou 450046, China; Yan000zhang@163.com (Y.Z.); 15882217509@163.com (L.Z.); 13598090706@163.com (L.Y.); 18482109281@163.com (L.J.); liguanhui2175@163.com (G.L.); 15738858329@163.com (M.J.)

**Keywords:** wheat, *CAT*, gene family, genome-wide, expression pattern, abiotic stress

## Abstract

Catalases (*CATs*) are present in almost all living organisms and play important roles in plant development and response to various stresses. However, there is relatively little information on *CAT* genes in wheat and related *Triticeae* species. A few studies on *CAT* family genes in wheat have been reported. In this study, ten *CAT* proteins (*TaCATs*) were identified in wheat and classified into three groups based on their phylogenetic features and sequence analysis. The analysis of the structure and motif composition of the *TaCAT* proteins suggested that a segmental duplication event occurred in the *TaCAT* gene family. Collinearity relationship analysis among different species showed that there were three orthologous *CAT* genes in rice and in maize. By analyzing the *cis*-elements in the promoter regions, we speculated that *TaCAT* genes expression might be regulated by light, oxygen deficit, methyl jasmonate and abscisic acid, and by transcription factors such as MYB. A Gene Ontology (GO)-based analysis showed that *TaCAT* proteins may be related to the response to various stresses, are cytoplasm localized, and may function as antioxidant enzymes. RT-qPCR and transcriptome data analyses exhibited distinct expression patterns of *TaCAT* genes in different tissues and in response to various treatments. In this study, a comprehensive analysis of wheat *CAT* genes was performed, enriching our knowledge of *CAT* genes and providing a foundation for further functional analyses of this gene family in wheat.

## 1. Introduction

Plant growth and development are affected by external abiotic and biotic stresses [1]. Environmental stresses alter cellular redox homeostasis, leading to the overproduction of reactive oxygen species (ROS), such as hydrogen peroxide (H_2_O_2_), hydroxyl radicals (OH^−^), superoxide (O^2^•), and other oxygen radicals [2,3]. ROS act as signaling molecules mediating many biological processes, but excessive accumulation of ROS in living plants can lead to oxidative stress damage to cells [4,5]. To adapt to the living environment, these highly toxic ROS must be converted into less reactive forms. Plants have evolved effective detoxification mechanisms, including non-enzymatic and enzymatic detoxification systems, to protect themselves from oxidative damage [2]. H_2_O_2_ is a major ROS and serves as an important messenger in abscisic acid (ABA) [6] and stress response pathways [7,8]. As the major mechanisms for removing ROS, the enzymatic systems include various types of antioxidant enzymes. Catalases (CATs) are considered the most potent ROS scavengers because of their strong affinity for H_2_O_2_ [3].

CATs have been found in almost all living organisms [9]; they have been studied extensively [3]. Many studies have illustrated that plant *CAT* gene expression is involved in the regulation of growth, development, and response to environmental stimuli [10,11,12]. In Arabidopsis, *AtCAT* genes encode a small family of proteins, including *AtCAT1*, *AtCAT2* and *AtCAT3*, which catalyze the decomposition of H_2_O_2_ and play an important role in controlling ROS homeostasis [13]. *AtCAT1* expression is induced by ABA, which is mediated by MAPK cascades [14], but it does not seem to respond to circadian rhythms [15]. *AtCAT2* is mainly expressed in leaves; it can be induced by light and cold, and may also be regulated by the circadian clock [15]. It has been reported that an *AtCAT2* mutant (*cat2*), defective in *AtCAT2* expression and, exhibiting 20% of the wild-type leaf catalase activity, accumulates more H_2_O_2_ than the wild type under normal growth conditions [16]. *AtCAT3* is highly expressed in the whole plant at all developmental stages [17], is regulated by CPK8, and participates in ABA-mediated stomatal regulation in response to drought stress [6]. In rice (*Oryza sativa*), three *CAT* genes have been identified, including *OsCATA*, *OsCATB*, and *OsCATC*. Previous studies have demonstrated that *OsCATA* and *OsCATC* are the most stress-responsive members. The overexpression of *OsCATA* or *OsCATC* increases drought resistance in rice [8]. Moreover, OsCATC can be phosphorylated and activated by STRK1, improving both salt and oxidative tolerance in rice [10].

In addition, three *CAT* genes have been identified in tobacco (*Nicotiana tabacum*) [18]. Here, 3-AT (3-Amino-1,2,4-triazole) reduces catalase activity by affecting catalase mRNA abundance, whereas paraquat induces *NtCAT3* expression. In sweet potato (*Ipomoea batatas*), *SPCAT1* plays a role in H_2_O_2_ homeostasis in leaves and in the plant’s response to environmental stress [19]. Interestingly, ectopic expression of *CAT* can also affect CAT activity and affect plant resistance to adverse conditions. For example, the expression of a wheat catalase gene in rice can enhance tolerance to low temperatures [20]. In addition, the ectopic expression of maize *CAT2* (*ZmCAT2*) in tobacco can induce CAT activity, improving pathogen resistance [21].

Several recent studies have firmly established that the ROS-scavenging enzymes can be mediated by several microRNAs (miRNAs) to improve oxidative stress tolerance in plants [22,23]. miRNAs are a class of endogenous non-coding RNAs (approximately 22 nucleotides) that participate in plant development and responses to environmental stress by negatively regulating target gene expression [24,25]. Recent studies indicate that higher miR408 expression contributes to enhanced antioxidant capacity, and improve tolerance to salinity, cold, and oxidative stress in Arabidopsis [26]. Furthermore, miR166 has been found to be involved in Cd stress response through regulating its target gene *OsHB4* in rice [27].

Common wheat is the most widely cultivated crop on Earth, contributing about a fifth of the total calories consumed by humans [28]. In a previous study, ten *CAT* genes have been systemically identified in wheat. *TaCATs* are classified into three classes, and their potential role in wheat development and different stress conditions have been revealed [20,29]. In addition, the increasing evidence that catalase and H_2_O_2_ coupled with the flour b* colour variation [30]. Studies have confirmed that *TaCAT3-A1* is expressed throughout seed development and associated with flour b* color variation rather than *TaCAT3-A2* [30,31]. In contrast to the research progress in other species, knowledge of the *CAT* genes in wheat is still limited [20]. The release of wheat whole-genome sequence data [28] enables systematic identification and analysis of wheat *CAT* genes at the genomic level. Therefore, in this study, a comprehensive genome-wide analysis of *CAT* genes in wheat was carried out. In addition to the phylogenetic relationships, gene structures, gene locations, conserved domains, *cis*-elements in promoters, and collinearity among species of *TaCATs* were investigated; in addition, the evolution history, gene duplication, functional annotations, the miRNA target sites of *TaCAT* genes, and the subcellular localization of TaCAT2-A/B protein have also been analyzed. Relative expression levels of *TaCAT* genes showed remarkable changes in response to different stress treatments and ABA. The results increase our understanding of the evolutionary history and biological function of *TaCAT* genes in wheat. This study also paves the way for further investigation into the function of *CAT* gene families.

## 2. Results

### 2.1. Identification of the CAT Gene Family in Wheat

A genome-wide screening of wheat proteins using HMMER v3.1 utilizing the Hidden Markov Model profile of the catalase domain (PF00199) specific for the *CAT* family as query revealed 19 candidates. These proteins were further verified using the NCBI Batch CD-search and Pfam. Proteins without a catalase domain were removed, and ten *TaCAT* genes were identified (Table 1).

Identified *TaCAT* genes were named as *TaCAT1-A*, *TaCAT1-B*, *TaCAT1-D*, *TaCAT2-A*, *TaCAT2-B*, *TaCAT2-D*, *TaCAT3-A1*, *TaCAT3-A2*, *TaCAT3-B*, and *TaCAT3-U* according to the wheat gene symbolization guidelines (Table 1). Analysis results indicated that there were two alternative spliced forms of *TaCAT1-D*, *TaCAT2-A*, *TaCAT2-B*, and *TaCAT3-A1*, three alternative spliced forms of *TaCAT2-D*, and no splice variants of any of the rest. The coding sequence (CDS) lengths of these genes ranged from 873 to 1479 bp, whereas the deduced protein sequence lengths ranged from 290 to 494 aa, the pIs ranged between 6.17 (for TaCAT3-A2) and 6.78 (for TaCAT3-B), and the molecular weight (MW) was between 33.51 and 56.92 kDa.

### 2.2. Phylogenetic Relationship and Sequence Analysis of TaCAT Genes

The phylogenetic relationship of *CAT* genes from wheat, Arabidopsis (dicot model plant), *Brassica. napus* (allopolyploid), rice (monocot model plant), sorghum (model C4 monocot), and maize (monocot model plant) was analyzed. A phylogenetic tree was presented in Figure 1. The tree showed that *CAT* genes of monocot plants could be classified into three groups (Groups I-III), and all the *CAT* genes of dicot could be classified into a three group. Similar results could be found in previous studies [29,32], which strongly support the reliability of the group classifications in our study.

The structure information of the *CAT* genes, that is, the exon-intron number and distributions in all 35 genomic sequences, was analyzed (Figure 2b). We found that the numbers of exons and introns were largely different between the genes from monocotyledons and dicotyledons. The number of exons differed from one to nine. The *CAT* genes of rice, sorghum, maize, and wheat possessed one to eight exons, whereas those from Arabidopsis and *B. napus* had seven to nine. In addition, a previous study suggested that an ancestral copy of the *CAT* gene had seven introns [33].

Conserved domains are functional units of proteins that are conserved across species [34]. To better understand the *CAT* gene family, the conserved domains of CAT proteins were analyzed using the NCBI Conserved Domain Database, and a schematic representation of the structure of all CAT proteins was constructed (Figure 2c). All CAT proteins contain one catalase core domain (PF00199, catalase) and one catalase immune-responsive domain (PF06628, catalase-rel). These two protein domains were highly conserved in wheat relative to those of previous studies done in cotton [35], *B. napus* [36], and rice [17]. *CAT* genes from different species shared high sequence and gene structure similarities, revealing evolutionary conservation among *CAT* gene family members.

### 2.3. Chromosomal Localization and Synteny Analyses of TaCAT Genes

To determine the physical position of *TaCAT* genes on chromosomes and gene expansion, we used the latest data from the wheat genome database IWGSC, which includes 97% of its sequences mapped to chromosomes as well as unanchored scaffolds that could not be mapped to any chromosome [28]. In total, 9 out of the 10 *TaCAT* genes were mapped onto wheat chromosomes (Figure 3); they were found to be unevenly distributed in the distal regions of the arms of eight different chromosomes. *TaCAT3-U* could not be incorporated into the physical map of any chromosomes. *TaCAT3-A1* and *TaCAT3-A2* were found to be clustered into tandem duplication event regions (Figure 3); apparently, these tandem genes had no homologous copies in any other subgenomes. Segment duplication events are generated after chromosome rearrangements and can also promote plant genome evolution [25]. To further infer the relationship between *TaCAT* gene family expansion and duplication, segment duplication events among *TaCAT* genes were also identified. The results showed that *TaCAT1-A*/*B*/*D* were homologous genes; similarly, *TaCAT2-A*/*B*/*D* and *TaCAT3-A1*/*B* constituted different homologous gene sets. No homologous gene of *TaCAT3-A2* was found in the other chromosomes. Common wheat originated after two naturally interspecific hybridization events; thus, each wheat gene usually has three homologous loci caused by polyploidization. These results indicate that the evolution of the *TaCAT* family was driven both by segment duplication and gene tandem events.

### 2.4. Conserved Motifs of TaCATs

Conserved domain analysis revealed that all the TaCAT proteins contained the catalase (PF00199) and catalase-rel (PF06628) domains (Figure 2). To further explore the potential functions of *TaCAT* genes, the conserved protein motifs of all transcripts of *TaCAT* were analyzed using the MEME tool (Appendix A). We found that motifs 1, 2, 3, 4, 5, 6, 7, 8, and 9 were conserved in most *TaCAT* transcripts, suggesting that these regions were important for CAT protein functions. Alternative splicing is a process in which the exons of RNA produced by the transcription of a major gene or mRNA precursor are reconnected by RNA splicing in a variety of ways. The translation of the resulting different mRNAs leads to different protein constructs; therefore, a single gene might code for multiple proteins. Alternative splicing events were found in some *TaCAT* genes; for example, *TaCAT1-D, TaCAT2-A*/*B*, and *TaCAT3-A1* had two transcripts and *TaCAT2-D* had three transcripts. In contrast with *TaCAT1-D.1*, transcript *TaCAT1-D.2* lacked motifs 7 and 9. Additionally, *TaCAT2-A.2* and *TaCAT2-D.2* lacked motif 9; *TaCAT2-B.1* and *TaCAT2-D.1* lacked motif 10; and *TaCAT3-A2* only contained motifs 1, 2, 3, 7, and 9.

### 2.5. Collinearity Relationship of CAT Genes among Different Plants

A collinearity relationship can provide insight into the evolutionary history of a genome [37]. To further investigate the collinearity relationship of *CAT* families in representative crop species, collinearity maps were constructed for wheat, maize, and rice (Figure 4). Interestingly, the results showed that a total of eight *TaCAT* genes had orthologous pairs in both maize and rice. Results indicated that *TaCAT1-A*/*B*/*D* might have had a common genetic origin with *OsCATC* (*Os03t0131200*) and *ZmCAT1* (*Zm00001d027511*); that *TaCAT2-A*/*B*/*D* might have had a common genetic origin with *OsCATA* (*Os02t0115700*) and *ZmCAT3* (*Zm00001d054044*); and that *TaCAT3-A1*/*B* might have had a common genetic origin with *OsCATB* (*Os06g0727200*) and *ZmCAT1* (*Zm00001d014848*). Our results suggest that these orthologous pairs may have already existed before the ancestral divergence of monocots.

### 2.6. Analysis of Cis-Elements in TaCAT Gene Promoters

To further understand *cis*-elements, the 2 kb 5*′* upstream region of the 10 *TaCAT* genes was analyzed using the PlantCARE database. The analysis results showed that, in addition to some basic core components, a series of elements, such as ABA-response element (ABRE), G-box (Sp1), MeJA-response element, and anoxia or anaerobic induction elements (Figure 5), were common to all *TaCAT* genes. These cis-elements are key components of abiotic stress responsiveness. For example, *TaCAT3-A1*/*A2* contained cis-elements associated with cold response, while *TaCAT1-B*/*D* and *TaCAT3-A2*/*B* contained salicylic acid (SA) response elements. The promoters of *TaCAT2-A*/*B*/*D* specifically contained an element related to meristem expression, indicating that these genes may be related to meristem development. In addition, some transcription factor binding sites, such as MYB binding sites, were found within most promoter regions. This suggests that *TaCAT* genes may be regulated by MYB transcription factors. In general, our results indicate that *TaCAT* genes of the same class may have different modes of action, and that genes of different classes may work together.

### 2.7. Functional Annotation Analysis of TaCAT Genes

Gene ontology (GO) has fueled the grand unification of biology, and the knowledge of the biological role of proteins in one organism often provides a strong inference of the role of similar proteins in other organisms [38]. To recognize the potential functions of *TaCAT* genes, a GO-based analysis was performed. According to this analysis, *TaCAT* genes were classified into three main categories: cellular component (CC), molecular function (MF), and biological process (BP) (Appendix A, Appendix A). The most significant MF GO term was catalase activity (GO: 0004096), consistent with the fact that CATs are antioxidant enzymes. In addition, the main CC GO terms were plasma membrane (GO: 0005886) and peroxisome (GO: 0005777), and the main BP GO terms were response to hydrogen peroxide (GO: 0042542), response to salt (GO: 1902074), circadian rhythm (GO: 0007623)*,* response to oxidative stress (GO: 0006979), response to bacterium (GO: 0009617), and response to ABA (GO: 0009737). In general, GO enrichment analysis results confirmed that, as antioxidant enzymes, *TaCAT* genes were involved in biotic and abiotic stress responses, ABA signal transduction, photoperiod, and appeared to be localized in the cytoplasm.

### 2.8. Expression Profiles of TaCAT Genes

Based on previously published transcriptome data, the expression patterns of the 10 *TaCAT* genes at different developmental stages (seedling, vegetative, reproductive and post-anthesis phases), tissues (roots, leaves/shoots, and spikes) and under different biotic and abiotic stress treatments were characterized (Figure 6). The analysis of the gene expression patterns revealed that those *TaCAT* genes belonging to the same subgroups shared similar expression patterns. *TaCAT2* showed a low expression after anthesis, while *TaCAT1*–*A*/*B*/*D* were highly and exclusively expressed in all stages of leaves development. In addition, *TaCAT3-A1*/*B*/*U* showed constitutive expressions patterns, that is, they were expressed in all different tissues at all stages. *TaCAT3-A1*/*B*/*U* were suppressed under cold and PEG treatments and induced under heat stress. *TaCAT2* were induced under heat treatment. All *TaCATs* were insensitive to CR. *TaCAT1* and *TaCAT2* were reduced under fusarium treatment. *TaCAT2* could be induced by powdery mildew and septoria treatments, and the expression of *TaCAT2* and *TaCAT3* were reduced under stripe rust. These data indicate that *TaCATs* may play important roles in all stages of wheat growth, development, and stress environment.

To further confirm whether the *TaCAT* genes responded to different abiotic stresses and ABA treatments, the mRNA expression patterns of the ten *TaCAT* genes from wheat seedlings subjected to NaCl, mannitol, heat (37 °C), cold (4 °C), and ABA treatments were analyzed. The results from RT-qPCR experiments showed that most *TaCAT* genes were significantly induced by most of the treatments (Figure 7). For example, under heat stress, the expression levels of *TaCAT3-A1*/*B*/*U* increased in a time-dependent manner, whereas those of *TaCAT2-A*/*B*/*D* initially increased and later decreased. In contrast, *TaCAT1*-*B*/*D* transcript levels decreased gradually. In the presence of ABA, *TaCAT* expression levels exhibited a slight decrease before increasing. Under NaCl and mannitol treatments, *TaCAT2-A*/*B*/*D* and *TaCAT3-A1*/*B*/*U* simultaneously showed a reduction in their expression levels, but they gradually increased as the treatment time increased. In contrast, *TaCAT1-A*/*B*/*D* showed a continuous downregulation under NaCl and mannitol treatments. The expression levels of *TaCAT2-A*/*B*/*D* were gradually but markedly increased in leaves under heat, NaCl, and ABA treatments, and peaked after 6 or 12 h of treatment, indicating that these genes might play a powerful role in ROS homeostasis under stress conditions. The different expression patterns of *TaCAT* genes under different treatments implied that *TaCATs* play different roles in response to stress conditions and signal transduction. Similar data were obtained using the glyceraldehyde 3-phosphate dehydrogenase (*GAPDH*) gene as another internal control, which was shown in Appendix A.

### 2.9. Subcellular Localization of TaCATs

To further understand the subcellular localization of TaCATs, the protoplast transformation method was used. TaCAT2-A/B-green fluorescent protein (GFP) fusion vectors driven by the 35S (CaMV *35S*) promoter were constructed. The empty vector 35S:GFP was used as a negative control, and the Arabidopsis gene *HY5* was used as a nuclear marker [39]. Plasmids were then co-transformed into Arabidopsis protoplasts. Laser scanning confocal microscopy results showed that the fluorescence signals of TaCAT2-A-GFP and TaCAT2-B-GFP fusion proteins were detected in both the nucleus and cytoplasm. Interestingly, distinct aggregated proteins were observed in the cytoplasm of cells that were transformed with the vector only (Figure 8). The fluorescence signals of the empty vector 35S:GFP were observed throughout the whole cell. Therefore, the TaCAT2-A and TaCAT2-B were confirmed to be localized in the nucleus and cytoplasm.

### 2.10. Analysis of MicroRNA (miRNA)Targeting Sites in TaCAT Genes

To investigate the relationship between miRNAs and *TaCAT* genes, the CDS of the *TaCAT* genes were used to search for putative target sites of miRNAs. As shown in Figure 9 and Appendix A, eight miRNA targeted seven *TaCAT* genes. *TaCAT1-A*/*B*/*D* were targeted by tae-miR395a/b at their 3*′* UTRs. *TaCAT2-A* exons were targeted by four miRNAs, including tae-miR408, tae-miR531, tae-miR9666a-3p, and tae-miR1137a, whereas tae-miR408 targeted two different sites of the gene. *TaCAT2-A* exons were targeted by tae-miR408 and tae-miR9666a-3p. Tae-miR5084 targeted *TaCAT3-A1* at its 3′ UTR region. *TaCAT3-U* was targeted by tae-miR5084 and tae-miR117. Our findings suggest that wheat miRNAs might regulate *TaCAT* gene expression by interacting with mRNA 3′ UTRs and exons. However, further research is needed to clarify the actual regulatory relationship between *TaCAT* genes and miRNAs.

## 3. Discussion

Plant *CAT* genes usually comprise a small gene family [32]. As the genomes of many species have been sequenced, genome-wide analyses of *CAT* families have been widely carried out. There are three *CAT* genes in Arabidopsis [15], maize [40], rice [11], cucumber [32], two *CAT* genes in *Hordeum vulgare* [41], seven in cotton [35], and ten *CAT* genes in wheat [36]. Understanding the biological functions of *TaCAT* genes and the molecular mechanisms underlying their responses to stressful environments are helpful for developing new wheat cultivars with enhanced resistance to multiple environmental challenges. However, the *CAT* family in wheat has not been thoroughly researched because of the complexity of the wheat genome.

In this study, a comprehensive genomic analysis of wheat led to the identification of ten *TaCAT* genes; they were symbolized based on their positions on wheat chromosomes (Table 1). Among the identified genes, nine were located on eight different chromosomes, while *TaCAT3-U* was provisionally mapped onto ChrUn (Figure 3). Gene tandem duplication and segmental duplication events are thought to be key mechanisms for increasing gene family diversity [29]. Gene tandem duplication events often occurred during plant evolution, leading to the expansion of gene families. If a chromosomal region within 200 kb contains two or more genes, the genes in this region are considered as the result of a tandem duplication event [30]. Interestingly, a tandem duplicated *TaCAT* gene pair (*TaCAT3-A1*, and *TaCAT3-A2*) was found in the wheat genome (Figure 3); these genes had no homologous copies in any other subgenome. In general, each wheat gene usually has three homologous copies in sub-genomes (A, B, and D) [42]. However, *TaCAT1-A*, *TaCAT1-B*, and *TaCAT1-D* were derived from 5A, 4B, and 4D, respectively. Previous studies confirm that this is because wheat A subgenomic progenitor had translocation that occurred between chromosomes 4AL and 5AL [29]. The results suggested that the *TaCAT* gene expansion was the result of genome polyploidization and gene duplications during evolution.

Gene structure has been identified as one of the representative traces of gene family evolution [43]. The exons and introns of *TaCAT* genes were found to be different between monocots and dicots. All *TaCAT* genes possessed one to seven introns and two to seven exons (Figure 2b). In addition, a previous study suggested that one ancestral copy of a *CAT* gene had seven introns [33]. Together, these results confirmed that *TaCAT* genes had gone through exon and intron changes during their evolution. Additionally, the corresponding TaCAT proteins contained catalase and catalase-rel domains (Figure 2c), suggesting that these genes were conserved during evolution.

Gene expression is regulated by the complex interaction of many *cis*-acting elements and trans-acting factors that participate in various pathways [32]. Previous studies have confirmed that some *CAT* genes could be induced by different treatments such as cold [13,33], ABA [6,34,35], drought [6,13,36], and light [37]. However, little research has been carried out to investigate the *cis*-acting elements of *TaCAT* gene promoters. Furthermore, a promoter analysis of the *TaCAT* genes revealed the presence of various stress-responsive elements, such as ABRE, and cold responsive, defense and stress response, and anaerobic induction elements (Figure 5). Some *TaCAT* promoters also included an MYB-binding site (MBS), suggesting that some *TaCATs* might be regulated by the MYB transcription factor. In a previous study, it was observed that ROS mediated the control of plant stem cell fate and was key to stem cell maintenance and differentiation in Ababidopsis [44]. Moreover, some elements of meristem expression, cell cycle regulation, light response, SA, MeJA, and auxin responses were also found in some *TaCAT* gene promoters. This suggests that *TaCAT* genes might also be involved in plant growth and cell differentiation by regulating ROS metabolism networks.

ROS have been widely studied as important signaling molecules and toxic components in plants [3,4,45,46,47,48]. Thus, the potential roles of *CAT* genes encoding catalases that control the homeostasis of ROS are of great importance. Studies have suggested that *CAT* family members are involved in the regulation of plant growth, plant development, and environmental stress responses [3,11,32,49]. For example, heterologous expression of a *TaCAT* gene in rice improves tolerance against low temperatures [20]. In the present study, we analyzed the transcript levels of *TaCAT* genes under heat, cold, NaCl, ABA, and mannitol treatments (Figure 7). The expression of all *TaCAT* genes changed dramatically when subjected to ABA treatment. These results indicated that *TaCATs* were closely related to the wheat ABA signaling pathway. Noteworthy, *TaCAT1-A*/*B*/*D* were found to be downregulated under NaCl and mannitol treatments, suggesting that they play specific roles when facing salt and drought stresses. We could not detect *TaCAT3-A2* expression in any of the samples examined. Nevertheless, the functional roles of *TaCAT* genes require further investigation.

In addition, valuable clues about the potential role of *TaCAT* genes in different tissues were obtained. The expression patterns of ten *TaCAT* genes were analyzed in different tissues at different developmental stages. The results demonstrated that *TaCAT3-A1*/*B*/*U* were constitutively expressed in almost all samples. *TaCAT1-A*/*B*/*D* showed a dramatically high level of expression in leaves, but it had a low expression level in the shoot axis at the milk grain stage (Figure 6). Furthermore, the expression of *TaCAT2-A*/*B*/*D* was high in leaves at the reproductive phase. In agreement with studies in other species [11,12,19,35,36,49], this study showed that some *TaCAT* genes were differentially expressed under various treatments, implying that *TaCAT* genes might be involved in environmental adaptation.

Subcellular location is an important biological characteristic of proteins [39], and it is quite useful to understand the mechanisms underlying protein cellular activities. The catalases of Arabidopsis were found to be localized in the peroxisome (the main site of H*_2_*O*_2_* production), whereas the rice catalases were predicted to be localized both in the cytoplasm and peroxisome [17]. Subcellular localization analyses showed that TaCAT2-A/B-GFP fusion proteins were localized in both the nucleus and cytoplasm, and that they formed protein aggregates in the cytoplasm (Figure 8). The formation of these aggregates and their molecular functions requires further study.

miRNAs are single-stranded, noncoding RNAs that post-transcriptionally regulate target gene expression in plants and animals [50,51]. miRNAs participate in plant development and other physiological processes [52]. Previous studies have confirmed that *CAT* genes of rapeseed [36] and cotton [35] were targets of some miRNAs. An analysis of putative miRNA-targeting sites in the *TaCAT* genes showed that seven of these genes might be regulated by up to eight different miRNAs. Two members of the tae-miR395 family targeting three *TaCATs* (*TaCAT1-A*/*B*/*D*), while miR408 targeting two *TaCATs* (*TaCAT2-A*/*B*). miR395 has been reported to involve in cadmium detoxification in *B. napus* [53], resistance to leaf spot disease in apple [54], and upregulated under sulphate and phosphate (Pi) deficiency in Arabdopsis. miR408 has been reported to have pleiotropic effects on plant growth such as seed yield and leaf area in Arabdopsis [55], regulating heading time in wheat [56]. These studies suggest that these tae-miRNAs might play important roles against different stresses as well as participate in plant development by regulating the transcript level of the *CAT* genes in wheat.

## 4. Materials and Methods

### 4.1. Identification and Characterization Analysis of TaCAT Genes

The nucleotide, protein sequence, and gene annotation (General Feature Format Version 3, GFF) files of *Triticum aestivum* were downloaded from the public database IWGSC (http://www.wheatgenome.org/ accessed on 20 April 2021) and the specific catalase (PF00199) Hidden Markov Model was downloaded from the Pfam database (http://pfam.sanger.ac.uk/ accessed on 23 March 2021). The HMMER 3.0 [57] search tool was used to screen the wheat CAT proteins, and 19 candidates were originally obtained. These 19 protein sequences were further identified using the NCBI Conserved Domain database (https://www.ncbi.nlm.nih.gov/Structure/cdd/ accessed on 23 March 2021) and Pfam database (http://pfam.xfam.org/ accessed on 23 March 2021). After removing the redundant sequences, 10 CAT proteins were obtained. These 10 *CAT* genes were named as the wheat gene symbolization guidelines (http://wheat.pw.usda.gov/ggpages/wgc/98/Intro.htm/ accessed on 23 March 2021) and selected for further studies.

The molecular weight (MW) and theoretical isoelectric point (pI) were computed using the ExPASY Compute pI/Mw tool (https://web.expasy.org/compute_pi/ accessed on 23 March 2021). *TaCAT* positions on chromosomes and orthologous genes in rice, *Arabidopsis*, and maize were identified using online tools (http://wheat.cau.edu.cn/TGT/ accessed on 23 March 2021). The subcellular localization of TaCAT proteins was predicted using the PSORT: protein subcellular localization prediction tool (https://www.genscript.com/psort.html/ accessed on 23 March 2021).

### 4.2. Phylogenetic Analysis of TaCAT Genes

To further clarify the evolutionary relationship between TaCATs and CATs of other species, CAT protein sequences of *Arabidopsis*, maize, sorghum, *B. napus*, and rice were downloaded and compared. Full-length protein alignments were performed using MUSCLE [58], and a phylogenetic tree was constructed using MEGA-X software (https://www.megasoftware.net/ accessed on 20 December 2021) using the neighbor-joining method with 1000 bootstrap replications. The tree was further modified using the Evolview software (https://evolgenius.info//evolview-v2/ accessed on 20 December 2021).

### 4.3. Chromosomal Localization and Collinearity of TaCAT Genes

By analyzing the GFF3 genome annotation files of wheat, *TaCAT* genes were mapped to the chromosomes, and the gene density profile of each chromosome was also obtained. The MCScanX toolkit program was used to study gene duplication and collinearity. Subsequently, we utilized TBtools software to visualize chromosomal localizations and gene duplications within species and among different species.

### 4.4. Conserved Domains, Motifs, and Gene Structure of TaCAT Genes

The conserved domains present in TaCAT protein sequences were identified using the NCBI Conserved Domain Search (CD Search) tool (https://www.ncbi.nlm.nih.gov/Structure/cdd/ accessed on 24 March 2021). The intron–exon structures of *TaCAT* genes were examined using the GSDS 2.0 (http://gsds.gao-lab.org/ accessed on 24 March 2021) online tool. The *TaCAT* conserved domains were visualized using TBtools software [59]. The conserved protein motifs in *TaCAT* genes were identified using the MEME online software (https://meme-suite.org/meme/ accessed on 24 March 2021). The parameters were set to expect 10 motifs distributed in the sequences, and no repetition was allowed.

### 4.5. GO Enrichment Analysis of TaCAT Genes

The TGT website (http://wheat.cau.edu.cn/TGT/ accessed on 4 November 2021) was used to functionally annotate *TaCAT* genes using GO terms in three main categories (CC, BP, and MF). *p*-values were adjusted by the BH correction. A false discovery rate of 0.05 was applied to the filter. The number of genes in the background ranged from 5 to 1200.

### 4.6. Cis-Acting Element Analysis

To further identify the putative cis-regulatory elements of the promoter regions of the *TaCAT* genes, 2–kb 5′ upstream sequences of the *TaCAT* genes were obtained using Tbtools software. The various putative *cis*-regulatory elements of these sequences were further analyzed using PlantCARE databases (http://bioinformatics.psb.ugent.be/webtools/plantcare/html/ accessed on 4 November 2021). The diagram was visualized using TBtools software [59].

### 4.7. Plant Materials and Stress Treatments

Chinese Spring seeds were sterilized and germinated in a chamber under a photoperiod of 16 h light/8 h dark, 22 °C, 60% humidity. After 3 days, the seedlings were transferred into Hoagland liquid nutrient solution, and the solution was changed every two days. Seedlings at the three-leaf stage were exposed to different treatment conditions. For salt stress treatment, seedlings were transferred to Hoagland liquid nutrient solution containing 200 mM NaCl. For cold and heat stress treatments, the seedlings were kept in culture chambers at 4 °C and 37 °C, respectively. For drought and ABA treatments, seedlings were transferred to Hoagland liquid nutrient solution with 300 mM mannitol or 200 mM ABA, respectively. Samples were collected at 0, 1, 2, 6, 12, and 24 h after each treatment, immediately frozen with liquid nitrogen, and stored at −80 °C until further use.

### 4.8. RNA Extraction and Gene Expression Pattern Analysis

Total RNA was extracted from each sample using RNAiso Plus (Taraka, Code No: 9108, Kota, Indonesia). The quality of RNA was analyzed by agarose gel electrophoresis and quantized quantified using a Nanodrop spectrophotometer. A PrimeScript RT reagent Kit with gDNA Eraser (Perfect Real Time) (Taraka, Code No: RR047Q) was used to reverse transcribe RNA into cDNA. RT-qPCR analyses were carried out on a Roche Lightcyler480 instrument (Roche, Basel, Switzerland) using TB Green^®^ Premix Ex Taq™ II (Tli RNaseH Plus) (Taraka, Code No: RR820A). The *TaActin* and *TaGAPDH* gene were used as an internal control, and each *TaCAT* gene was analyzed with three technical replicates. A standard two-step PCR amplification procedure was performed as follows: 95 °C for 10 s, 60 °C for 10 s, and 72 °C for 20 s, 40 cycles. The cycle threshold (CT) value of the real-time PCR was further analyzed using the 2^−^^△△CT^ method. Time 0 h (for each treatment) was normalized to 1. The primers used in this study for RT-qPCR were designed using NCBI Primer-BLAST tool (https://www.ncbi.nlm.nih.gov/tools/primer-blast/ accessed on 1 December 2021) and are listed in Appendix A. The results of RT-qPCR were compared by one-way analysis of variance (ANOVA) followed by post-hoc Tukey’s test at *p* < 0.05.

### 4.9. Expression Analysis of the TaCAT Gene Family from RNA-Seq Data

To analyze the spatiotemporal expression patterns of *TaCATs*, the transcriptomic data of Chinese Spring, downloaded from the ExpVIP (http://www.wheat-expression.com/ accessed on 29 April 2021), were used for gene expression profiling; data from roots, leaves, and spike tissues at different developmental stages, different days post-anthesis, different tissues at the milk grain stages and under different stress treatments were included. Then, the expression levels of *TaCAT* genes were calculated and normalized with TPM (transcripts per million) values and heatmaps were generated using a TBtools heatmap illustrator tool [59].

### 4.10. Subcellular Localization Study of TaCAT Proteins

To generate the TaCAT2-A/B-GFP fusions under the control of the 35S promoter, the open reading frame (ORF) regions of the *TaCAT2-A/B* genes with homologous arms were amplified from the cDNA of Chinese Spring and cloned into an XbaI/BamHI-digested pAN580 vector using an NEBuilder HiFi DNA Assembly Master Mix (New England Biolabs, Beijing, China). The ORF of *HY5* (gene ID: AT5G11260) was amplified from the cDNA of Arabidopsis Col-0. The resulting 35S:TaCAT2-A/B-GFP and control vectors (35S:GFP, 35S:HY5-mCherry) were co-transformed into *Arabidopsis* protoplasts. The protoplast extraction and transformation steps were done as described previously [60]. The primers used in this study were listed in Appendix A.

The GFP fluorescence signals of the protoplasts were observed under a confocal microscope (Zeiss LSM710; Zeiss, Jena, Germany) at 488 nm for excitation and detected between 505 nm and 530 nm for emission. The mCherry fluorescence signals were observed using 552 nm and 600–650 nm as excitation and emission wavelengths, respectively. The chloroplast fluorescence signals were observed using 488 nm as excitation and 650–750 nm as emission wavelengths.

### 4.11. Prediction of miRNA Target Sites in TaCAT Genes

To predict the putative miRNA target sites, the CDSs of *TaCATs* were used to search for the complementary sequences from 119 published wheat miRNAs using the psRNATarget database (http://plantgrn.noble.org/psRNATarget/ accessed on 15 October 2021) with default parameters. All putative targeted sites of the *TaCAT* genes are shown in Figure 9.

## 5. Conclusions

Although *CAT* genes play important roles in all stages of wheat growth, development, and biotic/abiotic stress environment, the identification and molecular characterization of *CAT* genes in wheat is also unclear. In the present study, we performed a comprehensive analysis of the *CAT* gene family in wheat. We reported, for the first time, that *TaCAT*s gene expansion was the result of genome polyploidization and gene duplications during evolution. Meanwhile, we verified that TaCAT2-A and TaCAT2-B proteins were located both in the nucleus and cytoplasm. In addition, some *TaCAT* genes had miRNA binding sites that may be regulated by miRNA. These findings provide an understanding of the potential functional roles of *CAT* genes in wheat. However, the complexity of plant antioxidant systems raises many unresolved questions, and the potential regulatory mechanisms of *TaCAT* genes need to be further characterized.

## Figures and Tables

**Figure 1 ijms-23-00542-f001:**
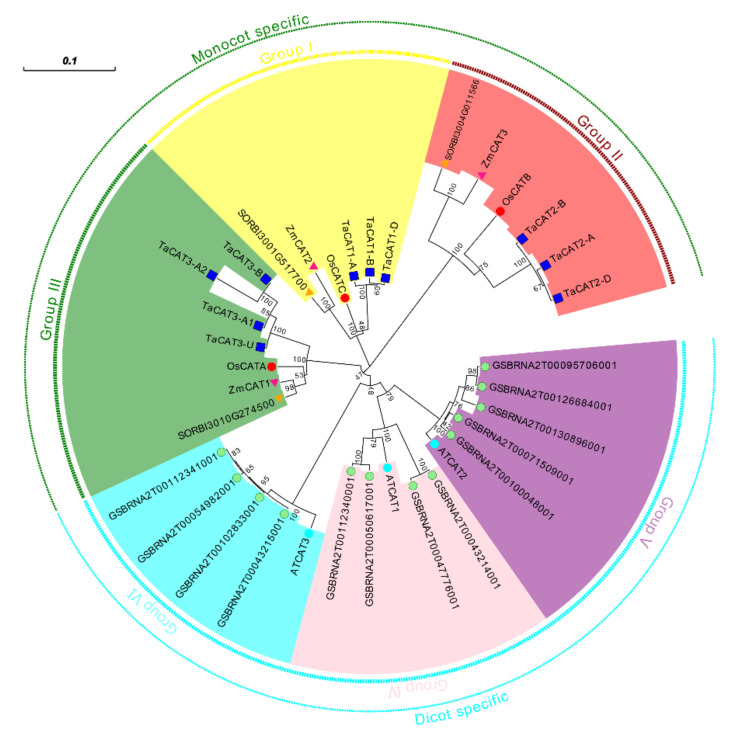
A neighbor-joining (NJ) phylogenetic tree of CATs in wheat (*Triticum aestivum* L. Ta), Arabidopsis (*Arabidopsis thaliana* L. At), maize (*Zea mays* L. Zm), rice (*Oryza sativa* L. Os), sorghum (*Sorghum bicolor* L.), and *B. napus* (*Brassica napus* L.). The tree was generated with the full-length CAT protein sequences of wheat (blue box), rice (red circle), Arabidopsis (cyan circle), maize (pink triangle), sorghum (yellow triangle), and *B. napus* (green circle). The phylogenetic tree showed six major phylogenetic groups and were indicated with different coloured backgrounds. Scale: 0.1.

**Figure 2 ijms-23-00542-f002:**
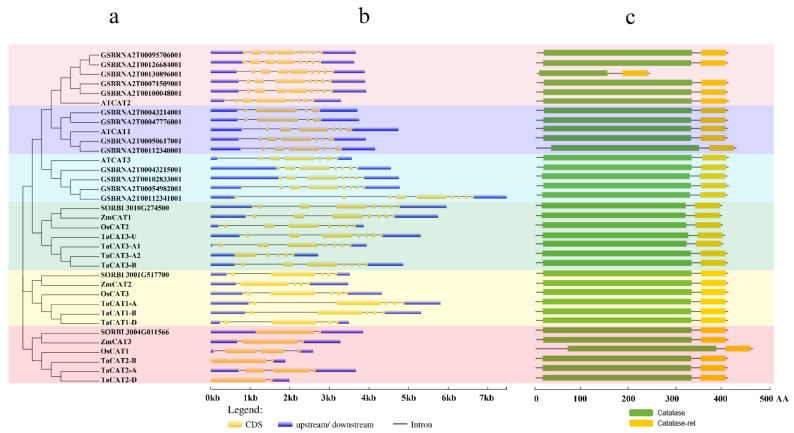
Phylogenetic relationships, gene structures, and conserved domains in *CAT* genes of Arabidopsis, *B. napus*, rice, sorghum, wheat, and maize. (**a**) Phylogenetic tree was constructed based on the full-length protein sequences of *CAT* genes using MEGA-X software with the calibration parameter of 1000; (**b**) exon–intron structure of *CAT* genes. The blue box represents the upstream/downstream sequences. The yellow boxes and the black horizontal lines represent the exons and introns, respectively; Scale indicates 1.0 kb; (**c**) schematic diagram of the conserved domain of CAT proteins. Catalase core domain (Catalase, PF00199) and catalase-related immune-responsive domain (Catalase-rel, PF06628) were shown in the blue and yellow boxes, respectively; Scale indicates 100 AA.

**Figure 3 ijms-23-00542-f003:**
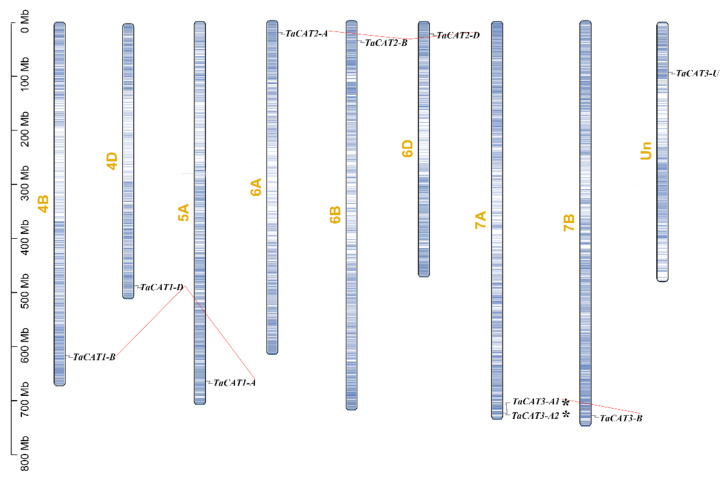
Schematic representations for chromosome distribution and duplication events of *TaCAT* genes. Chromosomal positions of Ten *TaCAT* genes were mapped on the basis of Wheat Genome (version IWGSC v1.0). Blue lines on each chromosome represented gene density. Segmental duplication events were indicated by red lines. * indicated the tandem duplication *TaCAT* gene pair. Only chromosomes that containing *TaCAT* genes were represented, and the chromosome number was indicated next to each chromosome.

**Figure 4 ijms-23-00542-f004:**
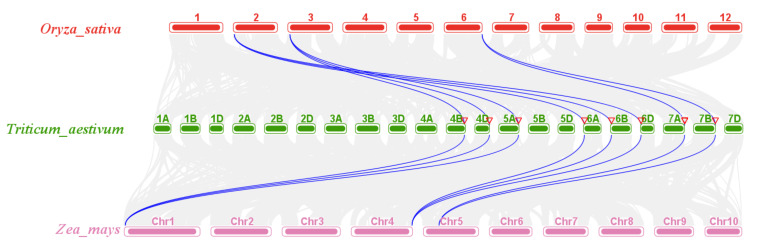
Synteny analysis of *CAT* genes between wheat, rice, and maize. Gray lines indicated all collinearity blocks within wheat and other plant genomes, and the duplicated *CAT* gene pairs were highlighted by blue lines. The positions of *TaCATs* were indicated with red triangles.

**Figure 5 ijms-23-00542-f005:**
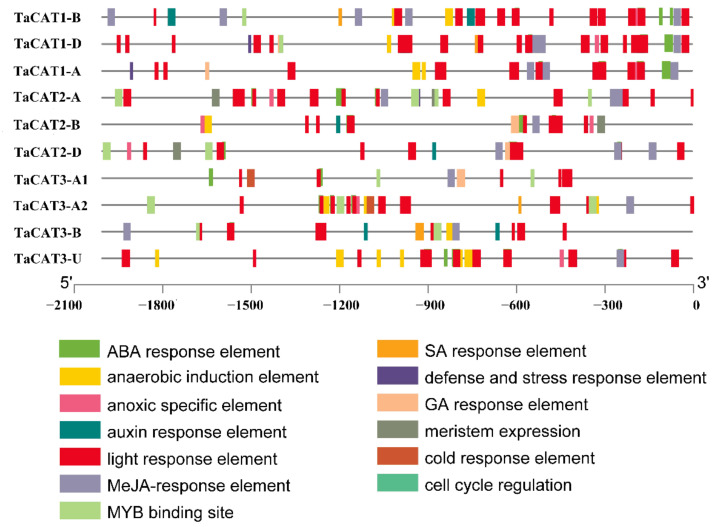
Analysis of promoter sequences of *TaCATs*. 2 kb 5′upstream region from the transcription start site (TSS) was selected and analyzed in silico using the Plantcare website. The analysis showed the presence of various cis-regulatory elements at the promoters of *TaCAT* genes. Different motifs were indicated by different color rectangles.

**Figure 6 ijms-23-00542-f006:**
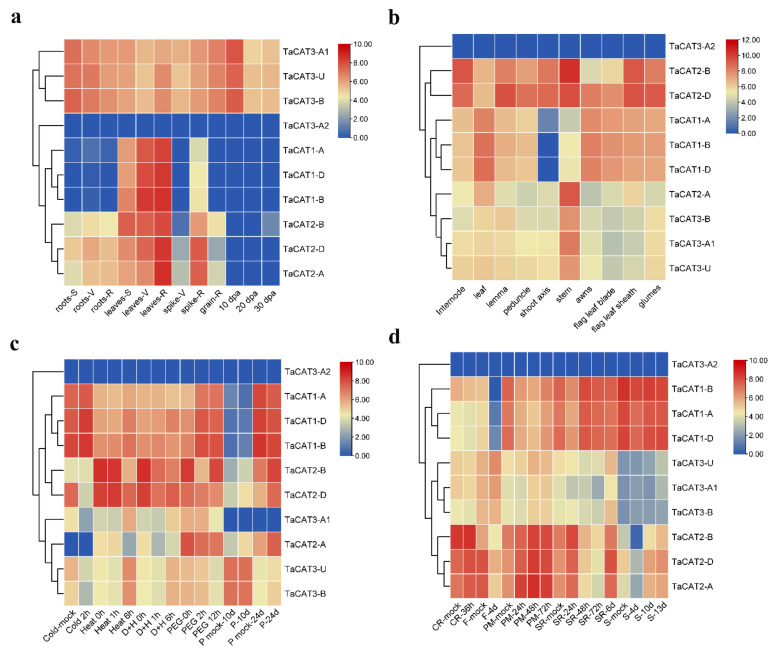
Expression analysis of *TaCAT* genes in different tissues of different developmental stages and stress treatments. (**a**) hierarchical clustering of expression profiles of wheat *CAT* genes in different tissues at seedling, vegetative, reproductive phase, and 10, 20, 30 dpa. S, seedling phase; V, vegetative phase; R, reproductive phase; dpa, day post-anthesis; (**b**) hierarchical clustering of expression profiles of *TaCAT* genes in different tissues at a milk grain stage; (**c**) hierarchical clustering of expression profiles of *TaCAT* genes under different abiotic stress treatments. D + H, drought and heat stress; P, phosphorus stress; (**d**) hierarchical clustering of expression profiles of *TaCAT* genes under different biotic stress treatments. CR, crown rot; F, fusarium hormones; PM, powdery mildew; SR, stripe rust; S, septoria.

**Figure 7 ijms-23-00542-f007:**
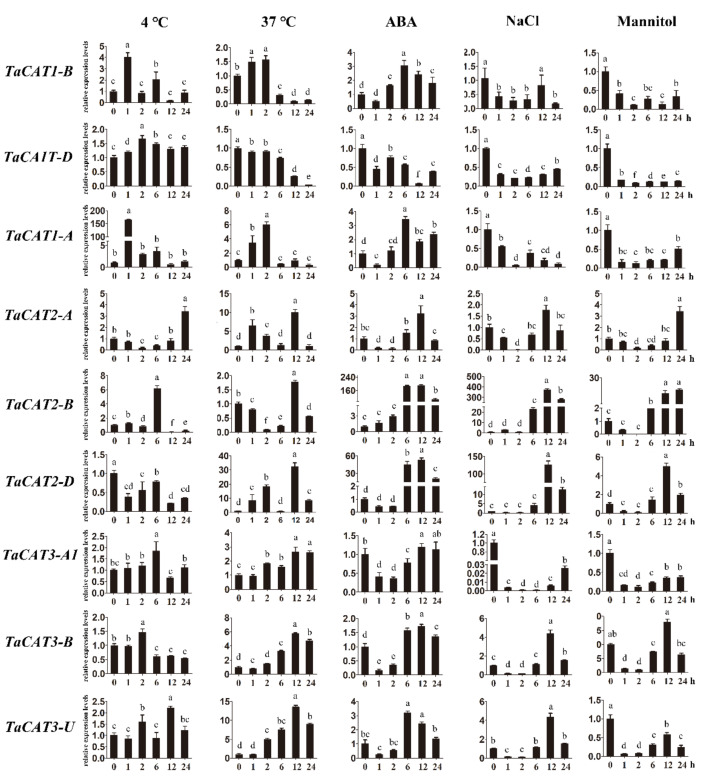
Expression analysis of *TaCAT* genes under different treatments by qRT-PCR. Relative expression levels of *TaCATs* in response to cold, heat, ABA, NaCl, and mannitol treatments for 0 h, 1 h, 2 h, 6 h, 12 h, and 24 h in the leaves at the three-leaf stage. Data were normalized with β-actin gene, and vertical bars indicate standard deviation error. Different letters indicate significant differences at *p* < 0.05 according to one-way ANOVA and post-hoc Tukey’s test.

**Figure 8 ijms-23-00542-f008:**
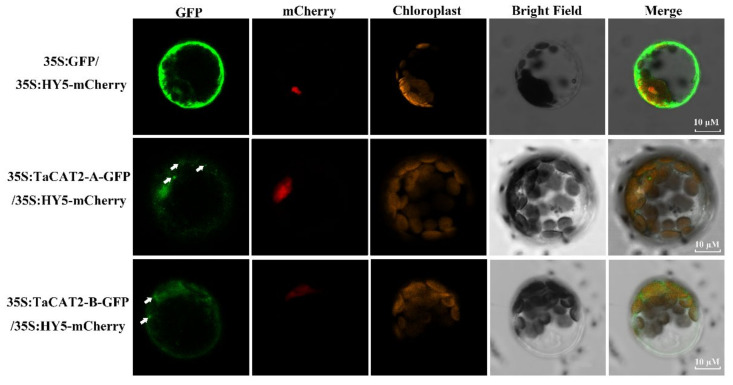
Subcellular localization analysis of TaCAT2-A and TaCAT2-B proteins in protoplast. TaCAT2-A-GFP and TaCAT2-B-GFP fusion protein were transiently expression driven by the 35S promoter in Arabidopsis protoplast. The fluorescence signals were observed under a laser scanning confocal microscopy. The green color represented the GFP fluorescence signals; the red and orange color represented the mCherry and the chloroplast fluorescence signals, respectively. The aggregated proteins were indicated by the white arrows. Scale bars were 10 μM.

**Figure 9 ijms-23-00542-f009:**
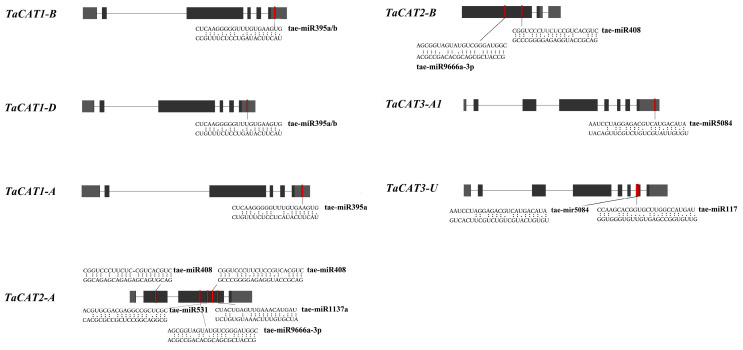
Analysis of miRNA targeting *TaCAT* genes in wheat. The heavy grey boxes represented the UTR regions and the black boxes were the exons. The miRNA complementary sites were shown in red boxes. The sequences of the miRNA and the complementary *TaCAT* sequences were shown in the expanded regions.

**Table 1 ijms-23-00542-t001:** Detailed information of identified *TaCAT* genes.

Gene Name	Gene ID	Chr	Position	CDS (bp)	Protein Length	pI	MW (KDa)	Description	Subcellular Localization	Orthologs
*TaCAT1-B*	TraesCS4B02G325800	4B	616987050-616991167(−)	1479	492	6.52	56.79	Catalase-1	Cytoplasm	*OsCATC*, *AtCAT1*, *ZmCAT2*
*TaCAT1-D*	TraesCS4D02G322700	4D	484657637-484661134(−)	1479	492	6.54	56.74	Catalase-1	Cytoplasm	*OsCATC*, *AtCAT1*, *ZmCAT2*
*TaCAT1-A*	TraesCS5A02G498000	5A	665466057-665470666(−)	1479	492	6.54	56.69	Catalase-1	Cytoplasm	*OsCATC*, *AtCAT1*, *ZmCAT2*
*TaCAT2-A*	TraesCS6A02G04170	6A	22021274-22023743(+)	1485	494	6.58	56.92	Catalase isozyme 2	Cytoplasm	*OsCATA*, *AtCAT1*, *ZmCAT3*
*TaCAT2-B*	TraesCS6B02G056800	6B	37250792-37253002(−)	1473	490	6.73	56.44	Catalase isozyme 2	Cytoplasm	*OsCATA, AtCAT1*, *ZmCAT3*
*TaCAT2-D*	TraesCS6D02G048300	6D	23075148-23077733(−)	1473	490	6.62	56.43	Catalase isozyme 2	Cytoplasm	*OsCATA, AtCAT2*, *ZmCAT3*
*TaCAT3-A1*	TraesCS7A02G549800	7A	724082577-724090076(+)	1479	492	6.66	56.49	Catalase isozyme 1	Cytoplasm	*OsCATB, AtCAT2*, *ZmCAT1*
*TaCAT3-A2*	TraesCS7A02G549900	7A	724113049-724114569(+)	873	290	6.17	33.51	Catalase isozyme 1	Cytoplasm	*OsCATB, AtCAT2*, *ZmCAT1*
*TaCAT3-B*	TraesCS7B02G473400	7B	730149989-730153654(+)	1479	492	6.78	56.53	Catalase isozyme 1	Cytoplasm	*OsCATB, AtCAT2*, *ZmCAT1*
*TaCAT3-U*	TraesCSU02G105300	Un	92716574-92720683(+)	1479	492	6.56	56.55	Catalase isozyme 1	Cytoplasm	*OsCATA, AtCAT2*, *ZmCAT1*

The gene name, gene ID, chromosomal position, coding DNA sequence (CDS), Isoelectric Point (PI), molecular weight (MW), description, sub-cellular localization, and orthologs of *TaCAT* genes.

## Data Availability

Not applicable.

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
