# Peer review of "Catalase (CAT) Gene Family in Wheat (Triticum aestivum L.): Evolution, Expression Pattern and Function Analysis"

_ijms, 2022, doi:10.3390/ijms23010542_

Round 1
Reviewer 1 Report
In this study, the authors identified CAT genes in wheat, analyzed the structure and motif composition and cis-elements in the promoter region, and investigated the genes' expression under different stresses. In the introduction, the authors did not talk about any progress about CAT genes in wheat. In the results part, a lot of sentences need to move to introduction or methods. I did not see the significance importance of the miRNA part, are they involved in regulation of the stress resistance? The authors need to do more work.Author Response
Please see the attachment.

Reviewer 2 Report
This is interesting and comprehensive genome-wide analysis of CAT genes in wheat.
The topic presents interest because increase the understanding of the evolutionary history and biological function of TaCAT genes in wheat and may also provide information for further investigation into the function of CAT gene families.
The abstract abides by all the editing instructions and presents very clear the objectives of the study. The introduction is well written and supported by well selected bibliographic data. All bibliographic sources are fairly recent and correctly mentioned in the text.
This is a good work. However, the following are recommended for improvement:
Some of the sentences are too long and could not convey any meaning. For example; the statement running from lines 100 to 104 should be restructured to bring out the meanings. If possible the statement can be broken down into smaller meaningful sentences.
In my opinion, the Discussion part is a bit limited, considering the abundance of results. The latest articles published on wheat, genome-wide analysis of CAT genes and other biochemical indicators in wheat should be introduced and correlated with the results obtained in this study.
I strongly recommend to the authors the following references:
- The Variability for the Biochemical Indicators at the Winter Wheat Assortment and Identifying the Sources with a High Antioxidant Activity. Plants 2021, 10(11), 2443.
2. Study on the yield and productivity elements of an assortment of winter wheat cultivated at ARDS Caracal. Annals of the University of Craiova - Agriculture, Montanology, Cadastre Series 2020, 50(1), 192-197.
Reviewer 3 Report
Dear Authors,
the manuscript presented for review is very well prepared. I have no substantive comments, I believe that the manuscript is refined and contains a number of results that achieve the purpose of the work as well as interesting, appropriate conclusions. My comments relate to minor editorial oversight:
Figure 2 - very small font, poorly legible
Figure 5 - legend - small font, poorly legible
Figure 7 - Signatures on the Y axis are practically impossible to read, on the X axis the numbers should be slightly larger
I believe that after these corrections the manuscript can be accepted for publication. However, these changes are important as they affect the readability and understanding of the manuscript.
Reviewer 4 Report
The authors have attempted to perform a bioinformatics analysis of Catalase gene family in Triticum aestivum L. Present manuscript entitled Catalase (CAT) gene family in wheat (Triticum aestivum L.): Genome-wide identification and expression in response to abiotic stress conditions lacked novelty. This manuscript has fatal flaws and cannot be published in its current state. As described in the comments below, the manuscript, particularly in the results parts, must be corrected properly and major English revision is required.
Introduction Line 70-72- In contrast to the research progress in other species, knowledge on the CAT genes in wheat is very limited, and few studies on this gene family have been reported [20]. Authors have not mentioned about the systemic analysis of catalase gene family in T. aestivum L. which has already been performed by Tyagi et al. 2021. Tyagi, S., et al., 2021. Molecular characterization revealed the role of catalases under abiotic and arsenic stress in bread wheat (Triticum aestivum L.). Journal of Hazardous Materials, 403, 123585. Authors have not mentioned about the three classes of catalases in the introduction.
Results Line-91-94- The nomenclature of TaCAT genes is not in accordance with the recommended rules for gene symbolization in wheat (http://wheat.pw.usda.gov/ggpages/wgc/98/Intro.htm.). This phylogeny seems incorrect, a total of three major groups of catalase has been reported in other findings. Please refer other articles such as Hu et al., 2016; Tyagi et al., 2021, etc. Also, bootstrap values and branch length are not mentioned on the phylogenetic tree which reduces its reliability.
“L. Hu, Y. Yang, L. Jiang, S. Liu, The catalase gene family in cucumber: Genome-wide identification and organization, Genet. Mol. Biol. 39 (2016) 408415. Authors have not analyzed the publically available transcriptome data of T. aestivum generated under abiotic stress conditions. However, the title of manuscript is focused around the expression of TaCAT genes in response to abiotic stress conditions
Why TaCAT is underlined in each sub-heading?
Line 271-273 Previous studies have shown that CAT genes are involved in environmental stress responses [11]. From example, heterologous expression of a TaCAT gene in rice improves tolerance against low temperatures [20]. Should be included in discussion.
Please avoid the overlapping of discussion in results section. Include statistical analysis to check the significance of qRT PCR results.
I suggest authors to include the functional characterization of differentially expressed TaCAT genes in Arabidopsis or some other heterologous system to bring novelty to this work.
Materials and methods Line 435- other species, CAT gene sequences of Arabidopsis It should be other species, CAT protein sequences of Arabidopsis. Line 469-470- Chinese Spring is an very important wheat landrace, that has been widely used in wheat genetics research. It should be included in discussion.
Conclusion Rewrite the conclusion. It seems like the overlapping of abstract.
Round 2
Reviewer 1 Report
No more comments.
Author Response
Thank you very much.
Reviewer 4 Report
Authors have revised the majority of comments, but the Ms still required a final proofreading and some other changes-
- What additional have been done in this study in comparison to the earlier report by Tyagi et al., 2021? That should be described in the introduction.
- Proposed gene name and gene model id should be the same to the earlier report by Tyagi et al to avoid any future complications.
- Why dicot catalases have formed a separate clade in phylogeny?
- Statistical method has not been described in the methodology.
- Why single TaActin has been used in RT-PCR, nowadays at least two reference genes are being used for proper expression analysis.
- I could see several typographical errors- like line 839- CAT should not be italics.
